# Potential Applications of Thyroid Hormone Derivatives in Obesity and Type 2 Diabetes: Focus on 3,5-Diiodothyronine (3,5-T2) in *Psammomys obesus* (Fat Sand Rat) Model

**DOI:** 10.3390/nu14153044

**Published:** 2022-07-25

**Authors:** Asma Bouazza, Roland Favier, Eric Fontaine, Xavier Leverve, Elhadj-Ahmed Koceir

**Affiliations:** 1Biology and Organisms Physiology Laboratory, Bioenergetics and Intermediary Metabolism Team, Nutrition and Dietetics in Human Pathologies Post Graduate School, University of Sciences and Technology Houari Boumediene, El Alia, Bab Ezzouar, Algiers 16123, Algeria; bouazza.asma@gmail.com; 2Laboratory of Fundamental and Applied Bioenergetics (LBFA), INSERM U1055, Univ. Grenoble Alpes, 16042 Grenoble, France; roland.favier@univ-grenoble-alpes.fr (R.F.); eric.fontaine@univ-grenoble-alpes.fr (E.F.); xavier.leverve@univ-grenoble-alpes.fr (X.L.)

**Keywords:** *Psammomys obesus*, 3,5-Diiodothyronine (3,5-T2), isolated hepatocytes, oxygen consumption, gluconeogenesis, ketogenesis, basal metabolic rate, type 2 diabetes, obesity

## Abstract

3,5-Diiodothyronine (3,5-T2) has been shown to exert pleiotropic beneficial effects. In this study we investigated whether 3,5-T2 prevent several energy metabolism disorders related to type 2 diabetes mellitus (T2DM) in gerbils diabetes-prone *P. obesus*. 157 male gerbils were randomly to Natural Diet (ND-controlled) or a HED (High-Energy Diet) divided in: HED- controlled, HED-3,5-T2 and HED- Placebo groups. 3,5-T2 has been tested at 25 µg dose and was administered under subcutaneous pellet implant during 10 weeks. Isolated hepatocytes were shortly incubated with 3,5-T2 at 10^−6^ M and 10^−9^ M dose in the presence energetic substrates. 3,5-T2 treatment reduce visceral adipose tissue, prevent the insulin resistance, attenuated hyperglycemia, dyslipidemia, and reversed liver steatosis in diabetes *P. obesus*. 3,5-T2 decreased gluconeogenesis, increased ketogenesis and enhanced respiration capacity. 3,5-T2 potentiates redox and phosphate potential both in cytosol and mitochondrial compartment. The use of 3,5-T2 as a natural therapeutic means to regulate cellular energy metabolism. We suggest that 3,5-T2 may help improve the deleterious course of obesity and T2DM, but cannot replace medical treatment.

## 1. Introduction

For a long time, it was accepted that natural thyroid hormone derivatives (THD) were considered an inactive hormone. However, literature data published over the past three decades gives a new renewed interest to THD as therapeutic means [1]. Indeed, it has been clearly established that THD can to act non-genomic (non-nuclear) mechanisms and exhibits similarly thyroid hormones (TH) activities, such as 3,5,3′,5′-tétraiodothyronine or thyroxine (T4) or 3,3′,5-triiodo-L-thyronine (T3). It should be noted that T3 exerts its physiological effects via genomic pathway (direct action on nuclear transcription) through nuclear TH receptors. Conversely, it is described that non-genomic actions of THD are mediated by mitochondrial binding sites or/cell membrane or/triggered by their endogenous catabolites [2]. So, the therapeutic strategy via the THD treatment without causing hyperthyroidism seems showed a promising future and the prevention of its metabolic complications, particularly in obesity and type 2 diabetes [3]. Undeniably, it is known that obesity and type 2 diabetes mellitus (T2DM) are deeply correlated to thyroid dysfunction [4]. Among THD, essentially 3,5-diiodothyronine (3,5-T2) which is the most thyroid hormone investigated in several therapeutic field, as novel thyroid hormones [5]. According to recent studies, 3,5-T2 which exhibits pleiotropic physiological actions without inducing thyrotoxic effects by acting mainly on mitochondrial function and could thus reverse intra/extracellular metabolic disorders, tissue damages and mitochondrial redox/phosphate dysfunction [6]. 3,5-T2 is synthesized from T3 deiodination pathway by Type III 5-deiodinase occurs in peripheral tissues, mainly in the liver [7].

In this study, we used the gerbil *Psammomys obesus* (*P. obesus*) as a spontaneous relevant polygenic human model for nutritional induced obesity, Non-alcoholic fatty liver disease, type 2 diabetes mellitus and atherosclerosis disease [8]. When the *P. obesus* is submitted on the transition from the natural low-energy-dense diets (Chenopodiaceae family) to synthetic-chow diet (a substantial high-energy-dense diets compared to natural diet), it develops obesity and metabolic disturbances characteristics such as hyperinsulinemia, insulin resistance, dysglycemia with progressive b-cell failure, dyslipidemia, hepatocellular steatosis and T2DM [9]. Based on the beneficial effect of 3.5-T2, the aim of this study was to investigate whether 3,5-T2 might reverse the insulin resistance, liver gluconeogenesis and ketogenesis dysregulation, hepatic steatosis (NAFLD), disturbance of basal metabolic rate and cellular oxygen consumption linked to mitochondrial redox/phosphate disorders. Currently, to our knowledge there are no studies that have investigated the effects of 3,5-T2 as a therapeutic target on hepatic energy metabolism in the Psammomys model.

## 2. Materials and Methods

### 2.1. Experimental Protocol Design

This study involved 157 adult male desert gerbil’s *Psammomys obesus* (*P. obesus*), older than 9 months (100–140 g) were trapped in Algerian North-Western Sahara desert in the area of Beni-Abbes (30°7′ North latitude, 2°10′ West longitude). Gerbil’s *P. obesus* were trapped during all seasons between January and December 2020. The experienced *P. obesus* were unscathed from all diseases by an authorized veterinarian under the control of the Algerian National Research Arid Zones Center. Afterwards *P. obesus* were transferred to colony room under controlled temperature (23 ± 1 °C) and fluorescent illumination was supplied 12-h light-dark cycle. The all experimentation was undertaken on euthyroid *P. obesus* and was not submitted to therapeutic treatment or nutritional supplementation. Daily, body weight is measured and calorie intake was calculated from 500 g of HED (High-Energy Diet) placed in food containers. Weekly, *P. obesus* are punctured from the retro-orbital venous plexus and blood was collected to determine biochemical and hormonal parameters. The body mass index (BMI) of *P. obesus* was assessed by dividing the weight (g) by the height square (cm); the tail of animal was not taken into account for the BMI calculation. After 5 weeks of experiments, the animals were anesthetized by urethane dimethacrylate (sigma Aldrich, Saint-Quentin-Fallavier, France), and sacrificed by cervical dislocation. Blood was collected in heparinized tubes. All the experimental procedures were authorized by the Institutional Animal Care Committee of the Algerian Higher Education and Scientific Research (DGRSDT; http://www.dgrsdt.dz, accessed on 1 January 2010). The permits and ethical rules has been achieved according to the Executive Decree n°10–90 (10 March 2010) completing the Executive Decree n°04–82 (18 March 2004) of the Algerian Government, establishing the terms and approval modalities of animal welfare in animal facilities. The investigation conforms to the Guide for the Care and Use of Laboratory Animals [DHHS Publ. No. (NIH) 85-23, revised 1996, Office of Science and Health Reports, Bethesda, MD 20892, United States of America]. The experiments conform to the ‘European Convention for the Protection of Vertebrate Animals used for Experimental and other Scientific Purposes’ (Council of Europe No 123, Strasbourg 1985).

### 2.2. Diet Composition

After a two week-initial accommodation phase, *P. obesus* were separated in individual cages and 1st randomization allowed dividing *P. obesus* into two diet-experimental groups (Figure 1):(i)Group I submitted on natural diet (ND): Thirty two *P. obesus* are strictly maintained *ad libitum* on ND. This ND group represents the diet-control *P. obesus*. This ND is composed by halophilic plants of the Chenopodiaceae family (*Traganum nudatum*, *Salsola foetidia*, *Suaeda fructosa* and *Artriplex halimus* species), freshly harvested in their native biotope [10]. The *P. obesus* subsists primarily or exclusively of these plants whose sap has a very high salt concentration [11]. The ND is characterized by a low caloric value [12].(ii)Group submitted on HED before randomization: one hundred twenty five *P. obesus* are maintained on laboratory synthetic chow pellets (Carfil Quality, Beyntellus, Belgium; https://www.carfil.be, accessed on 1 January 1982). This synthetic diet is intended for all experimental rodents but it is considered to be high-caloric diet (3.25 kcal/g of diet) compared to ND (0.40 kcal/g of diet) and hence, it is used to develop obesity and diabetes [13]. The HED contain: 47.4% carbohydrates, 25% proteins, 7.5% fat, with high energy carbohydrates (33.5% starch and 13.5% total sugar), fatty substances, vitamins, minerals. *P. obesus* HED-fed group are randomized into three subgroups according experimental protocol described in Figure 1: Group II or HED-controlled group: Thirty *P. obesus* continue to feed on HED without treatment or supplementation; Group III or 3,5-T2-treated group: Sixty-five *P. obesus* continue to feed on HED and submitted to a continuous and constant administration 3,5-T2 (3,5-diiodothyronine) pellet implant according subcutaneous technique (25 µg/100 g body weight/day) during 5 weeks. Group IV or Placebo-controlled group: Fifteen *P. obesus* continue to feed on HED and submitted to a continuous and constant administration of vehicle (isotonic saline solution 0.9% NaCl) according subcutaneous a placebo pellet implant. A placebo pellet was implanted at the same time as the *P. obesus* treated with 3,5-T2. In fact, 3,5-T2-treated group received the HED for 10 weeks and were subsequently treated for five weeks simultaneously with HED. The pharmacological dose of 25 µg 3,5-T2/100 g body weight (BW) was chosen after reviewing data from previous studies [14] which used 3,5-T2 as treatment and we have adapted for our study in gerbil *P. obesus.* Indeed, recent studies have shown that chronic treatment during 4 weeks with doses of 50 µg of 3,5-T2/100 g BW to male Rats HED pre-fed did not result in any thyrotoxic effect that might be of clinical relevance.
Figure 1Experimental protocol design in desert gerbils *Psammomys obesus*.
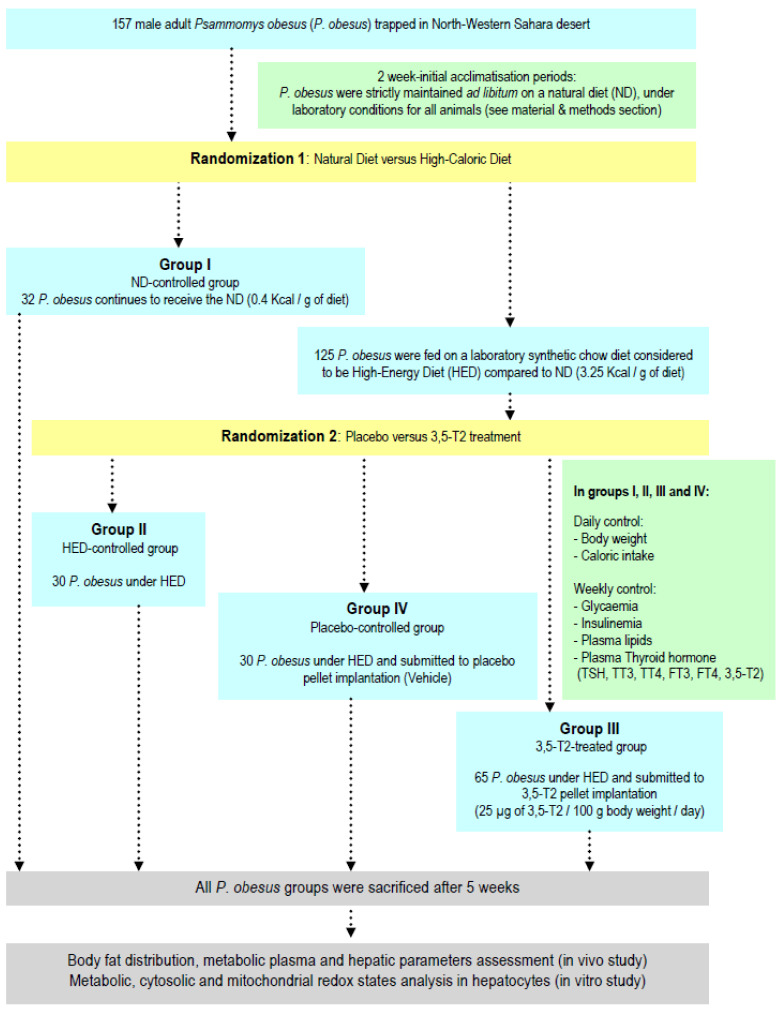



### 2.3. 3,5-Diiodothyronine Pellet Implantation Method

The pellet implantation technique has been described previously [15]. *P. obesus* were anesthetized by intraperitoneal injection with a simultaneous mixture of two anesthetic products: ketamine (Fort Dodge Animal Health Ltd., Southampton, UK) and diazepam (Roche Ltd., Kent, UK) in a ratio of 1:4. One mg/kg BW of Ketamine (Vetalar) and 1 mg/kg BW of diazepam (Valium) were administered. Anaesthesia was maintained lasts 1 h. In order to maintain body temperature during the surgery, gerbils were placed on a warm blanket. After interscapular shaving, a small incision of 4–5 mm of the skin allows the subcutaneous implantation of a small pellet (2.5 mm diameter) with containing 3,5-T2 or placebo with a stainless steel outer guide cannulae (10-gauge precision trochar) using a Kopf stereotaxic instrument to avoid damaging the nerves. Surgical intervention only lasts 2 to 3 min. The skin was sutured over the wound using surgical thread. The 3,5-T2 pellets were manufactured by Innovative Research of America (Sarasota, FL, USA) are constituted of a biodegradable matrix that effectively the active product in the animal. The 3,5-T2 pellets were implanted in order to provide a continuous and constant drug delivery over 60 days sustained release. Food intake was determined by weighing the food cups 2 h, 4 h, and 6 h after 3,5-T2 pellets implantation. The pellets were sent to us graciously by the Laboratory of Fundamental and Applied Bioenergetics (LBFA), INSERM, U1055, Grenoble, France.

### 2.4. Oxygen Consumption, Energy Expenditure, Basal Metabolic Rate Measurement

Energy expenditure (EE) as well as the nature of substrate oxidized was investigated by indirect calorimetry consists of cages, pumps flow controllers, valves, and analyzers. Between day 0 and day 7, the *P. obesus* were placed in an indirect calorimeter system (Oxymax, Columbus Instruments, OH, United States of America) for a period of 24 h at a room temperature of 28 °C. This system is computer-controlled in order to sequentially measure oxygen consumption (VO_2_) and carbon dioxide production (VCO_2_) as well as air flow in separate cages allowing simultaneous determinations. *P. obesus* are isolated in metabolic cages and room air is used as a reference to monitor ambient O_2_ and CO_2_ levels periodically. The system settings included a flow rate of 2 L/min, a sample line-purge time of 1.5 min. At predefined intervals, the computer sends a signal to store differential CO_2_ and O_2_ concentrations, flow rate, allowing computing VCO_2_, VO_2_, RQ (Respiratory quotient), and EE with data acquisition hardware (Metabolism, Panlab, Barcelona, Spain). The measuring system of VO_2_ and VCO_2_ allows calculating basal metabolic rate (BMR) or resting metabolic rates during which the rats were not moving according previous study for rodentia [16]. The contribution of protein catabolism to the RQ was assayed by measurement of urinary NPN (Non-protein nitrogen). Urine was collected in test tubes containing a sodium fluoride crystals and enough toluene to cover the urine completely. Filtered urine was frozen until analysis. To determine the thermal neutrality temperature, gerbils are fasted for 12 h and then placed individually in metabolic cages. Oxygen consumption and CO_2_ productions were continued for 8 hr and were determined every 5 min during the 30 min. Rectal temperature was measured with a thermometry. RQ is calculated according to the relation: RQ = VCO_2_/VO_2_ ratio. RQ is measured and used to calculate fat carbohydrates, lipids or proteins oxidation. EE was determined by RQ as previously described [17]. RQ ratio of 1.0 indicates exclusive carbohydrate oxidation while a ratio of 0.7 specifies lipid oxidation and ratio of 0.8 identifies protein oxidation.

### 2.5. Adipose Tissue Samples and Adiposity Index Estimation

At the end of the experimental period (5 weeks), assessment of body fat distribution content in *P. obesus* was made by surgically removing and weighing selected major adipose tissue depots from animals after sacrifice. These fat depots included the visceral adipose tissue witch integrated the retroperitoneal, omental and mesenteric tissue. Retroperitoneal fat pad was taken as the distinct depot behind each kidney along the lumbar muscles such as perirenal, Intramuscular and suprascapular. Epididymal fat consisted of adipose tissue on top of the epididymis. Subcutaneous adipose tissue depot is predominantly located in abdominal area and between the muscle and skin, essentially on top of the hind legs (inguinal depot) and upper back (dorsal depot). Brown adipose tissue has been dissected in the intrascapular area. The adiposity index was adapted according previous study in rats [18]. This data point was utilized to confirm obesity in the gerbil’s *P. obesus*. Obesity was defined based on the adiposity index; and the degree of adiposity was evaluated using nature body fat repartition. In our study, the most important white adipose tissues are the visceral, epididymal, subcutaneous and suprascapular. Brown adipose tissue is not considered in the adiposity index calculation.

### 2.6. Plasma and Hepatic Biochemical Analysis

All gerbil’s *P. obesus* groups were deprived of food for 12 h but free access to water was permitted before to beginning of the biochemical analysis. The blood samples from *P. obesus* were centrifuged at 3000 rpm for 10 min, and serum or plasma was obtained. Fasting samples were immediately put on ice and kept frozen at −80 °C until analyses were performed. Serum glucose, lipids profile [total cholesterol (TC), triglycerides (TG) and ketone bodies (3-βhydroxybutyrate + acetoacetate) were measured using the enzymatic methods (Cobas Integra 400, Roche Diagnostics, Meylan, France). ALT (alanine aminotransferase) plasma levels were determined by Cobas^®^ automaton (Roche Diagnostics, Meylan, France). ALT concentration is a consequence of hepatocyte damage [19] and the degree of liver steatosis is positively correlated with the liver function parameter ALT and visceral adipose tissue [20]. Plasma insulin was measured using a double-antibody solid phase radioimmunoassay (RIA-Insulin-Cis bio kit, France). Insulin resistance was estimated by the homeostasis model assessment of insulin resistance (HOMA-IR) method applied to experimental investigations. HOMA IR = fasting glucose (mmol/L) × fasting insulin (mU/L)/22.5 [21]. Plasma glycosylated haemoglobin (HbA1C) was measured by turbidimetry (Roche Diagnostic Systems, Basel, Switzerland). Plasma non-esterified fatty acids (NEFA) were extracted by solid-phase technical extraction [22] and were determined by microfluorimetry method (Roche, Lörrach, Germany).

### 2.7. Plasma Thyroid Hormones Assays

TSH (Thyreostimulating hormone); TT4 (total Thyroxine); TT3 (total 3,5,3′-triiodothyronine); FT4 (Free Thyroxine or Free 3,5,3′,5′-tetraiodothyronine); FT3 (Free 3,5,3′-triiodothyronine) and TT2 (total diiodothyronine or 3,5-T2) were measured by radioimmunoassay (RIAs) with rodent standard according to the manufacturer’s protocol. The assay kits have been imported respectively from Amersham Biosciences, Buckinghamshire, England; RIAs FT4-immunotech, Marseille, France and RIAs FT3-CIS Bio International, Saclay, France. RIAs assay kits were used to quantify 3,5-T2 have been imported from Biocode Hycel, Liège, Belgium. The 3,5-T2 assay employed a specific 3,5-T2 antibody with 0.05% cross reactivity with T4 [23]. It should be noted that the reference values of plasma thyroid hormones are not homogeneous in the literature. They are very disparate between mass-spectrometry (MS) or HPLC coupled to tandem MS and previous RIAs method. The thyroid hormones normal ranges are lower with the recent LC-MS/MS [24] compared to RIAs method [25]. In our study, we have used the references range according to the supplier assay kits who allowed us to discuss our results (see Section 3).

### 2.8. Hepatocytes Isolation Method

16 h-fasted *P. obesus* were anesthetized by intraperitoneal injection of sodium pentobarbital (5 mg/100 g), and hepatocytes were next isolated according to the method of Berry and Friend [26], as modified from Groen [27]. Hepatocytes were resuspended in Krebs-Ringer bicarbonate buffer (120 mM NaCl, 4.8 mM KCl, 1.2 mM KH_2_PO_4_, 1.2 mM MgSO_4_, 24 mM NaHCO_3_, 2.4 mM CaCl_2_, 2% BSA, pH 7.4), saturated with O_2_/CO_2_ (95:5%). Using a shaking water bath, isolated hepatocytes (10 mg dry cells per mL) were incubated for 30 min at 37 °C in closed vials containing 2.5 mL of oxygenated buffer with either alanine (20 mM) or lactate + pyruvate (10:1 mM) together with octanoate (2 mM), in the presence or absence of 3,5-diiodo-thyronine (Innovative Research of America, Sarasota, FL, United States of America) at 10^−6^ M and 10^−9^ M. Regarding 3,5-T2 in vitro hepatocytes incubation procedure, we have adapted the method described by Goglia and col. [28]. After 30 min, 350 μL of the cell suspension were deproteinized with HClO_4_ (5% final), then centrifuged at 13,500× *g* for 5 min.

### 2.9. Intracellular Metabolic Fluxes Analysis

The supernatants were neutralized with KOMO buffer (2 M KOH, 0.3 M MOPS) for subsequent assays of glucose, lactate, pyruvate and ketone bodies (3-hydroxybutyrate, acetoacetate) by spectrophotometry [29]. A second sample from the vial was used for the determination of nucleotide content in mitochondrial and cytosolic fractions after separation of cells by a quick spin across a silicone oil layer. All the enzyme reagents were purchased from Roche (Meylan, France). Lactate, octanoate, pyruvate and alanine were procured from Janssen Cosmetics (Aachen, Germany), and bovine serum albumin (BSA) from Sigma chemical Co. (St Louis, MI, USA).

### 2.10. Assessment of Hepatic Intracellular Intermediary Metabolites

Glucose 6-phosphate (G6P), phosphoenolpyruvate (PEP) and 3-phosphoglycerate (3PG) were measured in the neutralized total intracellular fraction cytosolic fraction. Metabolites were determined by enzymatic procedures with either spectrophotometric or fluorometric determinations of NADH as described previously [30]. The net flux of gluconeogenesis (*J* glucose, μmoles/min/g dry cells) was calculated from the concentration of glucose and the total cell content in the incubation medium. The ketogenesis flux measured in this study comes from the octanoate degradation which ensures the ketone bodies production, and the glycolysis flux represents the lactate production comes from the alanine substrate. Cytosolic and mitochondrial adenylic nucleotides (ATP, ADP) were separated by high-performance liquid chromatography (Spherisorb 5 μm octadecyl silane 2, 18 cm, 4.6 mm) at 30 °C. Elution was performed with sodium pyrophosphate-pyrophosphoric acid (25 mM, pH 5.75), with a flow rate of 1.2 mL/min. Detection was performed spectrophotometrically at 254 nm [31].

### 2.11. Determination of Oxygen Consumption Rates

After incubation of hepatocytes (7.5 mg of dry cells·mL^−1^) in the presence or absence of 3,5-T2 as described above, the cell suspension with substrates was quickly saturated with O_2_/CO_2_ and immediately transferred into a shaking bath at 37 °C in closed vials containing 3.2 mL of Krebs Ringer bicarbonate calcium buffer. The oxygen consumption rate was determined by polarography method at 37 °C with a Clark electrode and expressed as natoms O_2_/min/mg protein.

### 2.12. Hepatic Protein, Glycogen and Lipids Determination

Aliquots of liver weighing from 300 to 350 mg were rinsed and homogenates with ice-cold physiological saline and used for chemical analysis. The protein content of hepatic samples was determined according to the Bradford method [32], using bovine serum albumin (BSA) solution as a standard. The hepatic content in triglycerides, esterified cholesterol and free cholesterol was carried out by a sequential quantitative method [33], and they were gravimetrically measured. Glycogen was extracted from tissue samples in KOH at 100 °C, and then assayed after acid hydrolysis according to previous method [34].

### 2.13. Statistical Analyses

All data are presented as mean ± standard deviation (SD). The Student’s *t*-test is adapted for analysis our data when two independent groups. Furthermore, we have used the one-way ANOVA extends the *t*-test to more than two groups for the comparison between the *P. obesus* group maintained on a natural diet (Group I) and *P. obesus* groups submitted on a laboratory synthetic-chow diet (II, III, and IV) for each parameter measured. On the other hand to compare 3,5-T2-treated group versus placebo group. Both methods are parametric and assume normality of the data and equality of variances across comparison groups. Pearson’s coefficient (r) correlation analysis was performed to quantify associations between the plasma levels of thyroid hormones profile, plasma and hepatic metabolic parameters and mitochondrial redox status. Statistical analyses were performed using SPSS 20.0 for windows (SPSS Inc., Chicago, IL, USA). The results were considered significant (* *p* < 0.05), very significant (** *p* < 0.01), highly significant (*** *p* < 0.001) or no significant (ns).

## 3. Results

### 3.1. In Vivo Effects of 3,5-T2 Treatment on Plasma Thyroid Hormone Profile

The experimental protocol described in Figure 1 specifies the in vivo steps of 3,5-diiodothyronine (3,5-T2) treatment at 25 µg dose/100 g body weight/day evaluated during 5 weeks. The experimental steps target the thyroid hormones-hypothalamic–pituitary–thyroid axis (HPT) dependent, and hepatic energetic metabolism during obesity and diabetes mellitus stage induced by HED (synthetic-chow diet) in male *P. obesus* gerbil. As shown in Table 1, the treatment by 3,5-T2 increases significantly plasma 3,5-T2 concentrations in group III compared to other groups. Besides, diet (ND or HED) and 3,5-T2 treatment did not influence thyroid weight. However, HED had an impact on thyroid mass (thyroid-to-body weight ratio) which decreased by 47%, 32% and 49% in groups II, III and IV compared to group I, respectively (*p* < 0.001). Interestingly, this ratio increases after 3,5-T2 treatment by 24% in group III versus placebo group (*p* < 0.001). Conversely, total 3,5,3′-triiodothyronine (TT3) levels were halved in the HED group (II, III and IV) versus natural diet group (*p* < 0.001). Nevertheless, 3,5-T2 treatment do not modify the TT3 levels versus placebo group. The FT3 levels are significantly reduced by 43% versus ND group and by 39% versus placebo group, respectively (*p* < 0.001). Interestingly, 3,5-T2 treatment drastically decreases the FT3/FT4-ratio, FT3/3,5-T2-ratio and FT4/3,5-T2-ratio by 39%, 68% and 48% respectively versus placebo group (*p* < 0.001). Interestingly, there was a significant positive correlation between FT3/3,5-T2-ratio and attenuation of insulin resistance (HOMA-IR index) in group III (r = +0.589, *p* < 0.0001). It should be noted that effects of 3,5-T2 treatment are dose-dependently manner, more marked at 10^−6^ M than 10^−9^ M.

### 3.2. In Vivo Effects of 3,5-T2 Treatment on Body Weight, Calorie Intake, Respiratory Quotient and Basal Metabolic Rate

As mentioned in Table 2, food consumption in *P. obesus* maintained on ND (as in natural desert environment) is very low. The daily energy intake was 31.5 ± 1.9 Kcal/100 g BW, which corresponds to about 76 g of halophilic plants/day. In the meantime, the daily energy intake in HED fed *P. obesus* was markedly increased (316 ± 11 Kcal/100 g BW), and their body weights were almost doubled as compared to ND group (group I). The *P. obesus* submitted on HED after 5 weeks gained significantly about 43% more weight than group I, at the same age, whether in the female or the male. The BMI values confirm that the HED gerbils groups are obese versus ND group (0.49 ± 0.01 vs. 0.38 ± 0.01 g/cm^2^, respectively, *p* < 0.001). These anthropometric parameters were diminished significantly in group III after chronic administration of 3,5-T2. Body weight and BMI was reduced by 23% and by 32%, respectively (*p* < 0.001) in group III compared to group IV (Table 2). Paradoxically, despite body weight loss, 3,5-T2 treatment increases calorie intake by 41% in group III versus group II but not modified in placebo group. Interestingly, 3,5-T2 treatment at 10^−6^ M and 10^−9^ M doses decreases significantly the respiratory quotient (RQ) in group III toward lipids utilization as energy substrate (RQ = 0.7) compared to groups I, II and IV which use carbohydrates as energy substrate (RQ = 1). There was a significant positive correlation between plasma triglycerides decrease and RQ value (r = +0.991, *p* < 0.0001) in group III. The basal metabolic rate (BMR) was significantly lower by 19% in ND-group than gerbils-untreated 3,5-T2 (group II), indicating greater energy expenditure under HED-diet. In addition, the 3,5-T2 treatment at 10^−6^ M dose increases significantly BMR by 27% in group III versus group IV, which explains the weight and BMI loss are associated to BMR (Table 2).

### 3.3. In Vivo Effects of 3,5-T2 Treatment on Body Adipose Tissue Distribution

As illustrated in Table 3, white fat repartition shows that visceral adipose tissue (VAT) and suprascapular adipose tissue (SAT) are most abundant in gerbil’s *P. obesus* under HED compared to ND. The VAT and SAT mass are significantly increased in group II reflecting the increase BMI in groups II compared to ND-controlled group (25.8 ± 4.32 and 12.5 ± 1.08 mg/g body weight, respectively, *p* < 0.001). In total VAT, the mesenteric adipose tissue is hypertrophied (51%) compared to omental and retroperitoneal fat (38% and 11%, respectively). Epididymal, subcutaneous, intramuscular, perirenal and gonadal adipose tissues are not influenced by the HED and are moderately developed compared to VAT and SAT (8.69 ± 1.47; 8.33 ± 1.18; 3.74 ± 0.27; 2.82 ± 1.61 and 1.93 ± 0.22, respectively). Adiposity index shows that group II is obese concomitantly to increase of BMI versus group I (Table 3). The 3,5-T2 hormonal therapy allowed to reduced significantly adiposity index in group III compared to group II et group IV (−72%). There was a significant positive correlation between plasma triglycerides attenuation and VAT loss (r = +0.785, *p* < 0.0001) in group III. The beneficial anti-obesity effect of 3,5-T2 treatment had a positive impact on VAT and SAT mass reduction (−61% and −12%, respectively). Besides, brown adipose tissue is increased by 40% under 3,5-T2 treatment in group III versus groups II and IV, but modestly versus group I (Table 3).

### 3.4. In Vivo Effects of 3,5-T2 Treatment on Plasma and Liver Metabolic Disorders

As shown in Table 2, when the *P. obesus* is fed to synthetic-chow diet (HED), it develops metabolic syndrome characterized by an insulin resistance (hyperinsulinemia, elevated HOMA-IR index), glucose intolerance (increased HbA1c levels) and plasma lipids disturb, and progressively chronic hyperglycaemia with ketoacidosis (group II). Dyslipidemia was especially due to plasma triglycerides and total cholesterol increase become paroxystic levels in group II versus group I (+80%, +81%, respectively). In contrast, on a halophilic plants diet (group I) gerbil’s *P. obesus* does not reveal impaired glucose tolerance, hyperinsulinemia or plasma lipid abnormalities. The 3,5-T2 administration largely lowered significantly plasma glucose (−74%, *p* < 0.001) in group III versus group II. Concomitantly 3,5-T2 treatment allowed to reduce the level of HbA1c hyperinsulinemia and HOMA-IR index in group III versus placebo group (−68%, −81%, −52%, respectively, *p* < 0.001). The glycaemia and HbA1c gradually declined throughout the treatment in group III until reaching a stabilized end-value near to group I (3.21 ± 0.51 mM, 17.9 ± 0.65 mM/mol, respectively). The reduction in adiposity could be reflected in the improvement of glycaemic homeostasis in HED-3,5-T2-treated. This indicates that abnormalities of sensitivity to insulin in gerbils group IV was fully reversed by 3,5-T2 treatment (Table 2) and the glucose-lowering 3,5-T2 effect was due to an inhibition of gluconeogenesis from lactate (see data below hepatocytes study). This highlights that 3,5-T2 treatment is able to reduces hyperglycemia. It should be noted that the HED increases lactacidemia in group III versus group I (+42%). Interestingly, the benefit of 3,5-T2 treatment allowed to attenuate significantly serum total cholesterol, but moderately serum triglycerides levels compared to group I (−69%, −48%, respectively), This powerful hypolipidemic effect by 3,5-T2 is associated to reduce serum lactate in group III to a level near to group I (0.73 ± 0.14 vs. 0.65 ± 0.09 mM, respectively). Conversely, 3,5-T2 treatment increased the serum non-esterified fatty acids (NEFA) and ketone bodies levels in group III versus placebo group (+32%, +52%, respectively). Taken together, improvements in these metabolic parameters revealed that 3,5-T2 treatment was able to reverse insulin resistance and an increase in fat burning to long-term HED-fed. On the over hand, the hepatic mass increase together with triglycerides accumulation was obviously indicative of severe liver deterioration in HED diet induced a substantial lipid-droplet accumulation (Table 3). This fact was supported by drastic increase of transaminases activity in ALT (Table 2) in groups II versus ND- controlled group (+64%). The plasma ALT level, well documented as a marker of hepatocyte damage, was significantly (*p* < 0.001) elevated in HED groupe II, whereas administration of 3,5-T2 to group III prevented this increase (23.9 ± 1.47 versus 81.3 ± 1.71 IU/L in placebo groups). This harmful total hepatic lipids accumulation was mitigated under 3,5-T2 treatment that actually normalized the liver mass-to-body weight ratio in group III versus placebo group (2.89 ± 0.33 vs. 3.64 ± 1.49% BW, respectively, *p* < 0.001). These results indicate that 3,5-T2 treatment can prevent hepatic steatosis. Concomitantly, the 3,5-T2 treatment allowed to decrease the hepatic glycogen content (−55%) in group III compared to the placebo group (Table 3).

### 3.5. In Vitro Effects of 3,5-T2 Treatment on Oxygen Consumption, Hepatic Gluconeogenesis, Ketogenesis, Intracellular Intermediary Metabolites, and Cellular Redox-Phosphate Potential

During fasting, oxygen consumption is significantly reduced in HED-controlled group compared to the ND-controlled group either from alanine (Ala) + Octanoate (Octa) or Lactate (L) + Pyruvate (P) + Octa (10.2 ± 0.84 vs. 19.4 ± 0.62 or 11.8 ± 0.47 vs. 20.1 ± 0.91 natoms oxygen/min/mg protein, respectively, *p* < 0.001). Interestingly, 3,5-T2 treatment increased significantly oxygen consumption under HED-diet in group III compared to placebo group either from Ala + Octa (Figure 2A) or L + P + Octa (Figure 2B). As shown in Table 4, the gluconeogenic rate was markedly higher in HED-controlled group as compared to ND-controlled group. The rate glucose production is increased by 48% from Ala + Octa (*p* < 0.001) and by 53% from L + P + Octa. It appears that lactate is an efficiency substrate than alanine in gerbils *P. obesus*. The active glycolysis in group II comparatively to ND group (3.27 ± 0.14 vs. 1.83 ± 0.12 µmol/min/g dry cells) is able to produce a significant amount of lactate. The 3,5-T2 treatment at 10^−6^ M decreased significantly hepatic glucose output from both gluconeogenic substrates conditions such as L + P + Octa (−47%, *p* < 0.001) or Ala + Octa (−43%, *p* < 0.001) in group III compared to placebo group. At the same time, 3,5-T2 treatment at 10^−6^ M reduced significantly the glycolytic flux from alanine by 48% (*p* < 0.001) in group III versus group IV (Table 4). This result explained the serum lactate depletion in group III treated in vivo with 3,5-T2 (Table 2). Of note, 3,5-T2 treatment failed to significantly inhibit hepatocytes gluconeogenesis in ND-placebo gerbils, yet the hepatocytes glycolytic pathway was activated but to a lesser extent than that in group III. The beneficial effect of 3,5-T2 on hyperlactacidemia had a positive impact on the significant decrease in cytosolic redox potential (NADH/NAD+) represented by the L/P ratio (Figure 3A). When next examining ketogenesis (Table 4), assessed by the initial oxidation rate of medium-chain octanoate fatty acid; 3,5-T2 treatment at 10^−6^ M strongly increase the hepatocytes ketone bodies synthesis in group III versus placebo group both from L + P + Octa or Ala + Oct (+44%, +37%, respectively, *p* < 0.001). The β-hydroxybutyrate (3-βOHB)/acetoacetate (AcAc) ratio is increased significantly in the group III versus placebo group (Figure 3B,C). Concomitantly, the effects of 3,5-T2 treatment on cytosolic and mitochondrial redox potentials have an impact on cytosolic and mitochondrial ATP production and phosphate potential (ATP/ADP). As shown in Figure 4, the treatment by 3,5-T2 allows to increase significantly the cytosolic ATP/ADP ratio (Figure 4A,B). Conversely, the 3,5-T2 treatment decreases significantly the mitochondrial phosphate potential in group III versus placebo group (Figure 4C,D). Regarding, intracellular intermediary metabolites (Table 4), we found 3,5-T2 treatment at 10^−6^ M increases significantly both phosphoenolpyruvate (PEP), 3-phosphoglycerate (3PG) and fructose 6-phosphate (F6P) levels in group III versus placebo group from L + P + Octa as gluconeogenic substrates (+37%, +21% and +18%, respectively, *p* < 0.001). At opposite, 3,5-T2 treatment decreases greatly glucose 6-phosphate (G6P) levels in group III versus placebo group (−46%, *p* < 0.001).

## 4. Discussion

Data presented in this study highlights 3,5-T2 has beneficial pleiotropic physiological actions. 3,5-T2 allowed to prevent the progression of insulin resistance side effects, regress weight gain, enhance liver gluconeogenesis and ketogenesis dysregulation, improve hepatic steatosis damage, re-establish the basal metabolic rate disturbance and restore the cellular oxygen consumption linked to mitochondrial redox/phosphate disorders. In our study, the 3,5-T2 administration for a period of 5 weeks does not influence the metabolism in gerbils *P. obesus* under natural diet (ND-controlled group). However, the effects of 3,5-T2 treatment were significantly observed in gerbils fed on High-Energy Diet (HED-controlled group) and to led some “therapeutic” effects [35]. Although the action mode of 3,5-T2 is classified as non-genomic (non-nuclear), it should be noted that some questions remain unanswered related to 3,5-T2 pathway acts via nuclear HT receptor (like T3) or whether it induces to rapid additional signals (plasma membrane, cytosolic compartment) to mediate changes on gene expression, or mitochondrial function to regulate energy metabolism. Actually, beneficial prospects for the use of 3,5-T2 as a therapeutic arsenal are derived from the recent novel development of a synthetic 3,5-T2 analog TRC150094, which might be used in obesity, diabetes and cardio-metabolic therapy application [36].

### 4.1. The First Point Is Linked to 3,5-T2 Treatment on Plasma Thyroid Hormone Profile

The studies undertaken on thyroid function in *P. obesus* model remain very limited and none studies on 3,5-T2 therapy. However, it is interesting to report some data on this model under natural diet composed by halophilic plants. In wild *P. obesus*, the thyroid gland is involved in adaptation to the desert life since the decrease in metabolic rate observed in desert animals, especially to related iodine, water and salt metabolism [37]. The plasma T4 production and T4 metabolic clearance were not different within *P. obesus* and other Gerbillidae species. However, plasma T3 levels and the thyroid ratio T3/T4 were significantly lower in *P. obesus* compared to other Gerbillidae species, such as Meriones, Dipodomys and jerboa. In addition, comparatively to Wistar rat, T4 and T3 levels in *P. obesus* were close to the inferior limits. Although the wild psammomys shows a euthyroid state, it seems that thyroid hormone profile in *P. obesus* is similarly to approach the Euthyroid Sick Syndrome [38]. Besides, it should be noted that plasma iodine levels were the same range for *P. obesus* and other Gerbillidae. Despite the elevated food iodine intake in *P. obesus*, the thyroidal function did not appear much modified. Indeed, the total iodine content of the thyroid in *P. obesus* was significantly higher compared to other Gerbillidae desert; nonetheless, thyroid weight is not significantly different per 100 g body weight [39]. It appears that *P. obesus* excretes the major part of the ingested iodine, concomitantly with salt renal excretion, as such manner that iodine and sodium coefficient assimilation being about 5%. Authors suggest that the low T3 levels in *P. obesus* are linked to high iodine intake which would favour T4 synthesis. This crucial regulation mode leads to reduced circulating thyroid hormone levels without stimulation of TSH secretion [40]. As emphasized previously, when *P. obesus* is fed a high calorie diet (HED) it develops the cardinal signs of diabetes mellitus, mainly impaired glucose tolerance. It has been reported that the thyroid hormone secretion interferes with other endocrine glands and establish major interactions, primarily with adenohypophysis, adrenal cortex and Langerhans islets. It is described that thyroid epithelium decrease significantly once diabetes has developed [41]. In our study, regarding TSH, T4 and T3 levels, in both *P. obesus* groups no showed substantial differences from the normal euthyroid values. Our study highlights that 3,5-T2 treatment does not alter the thyrotropic axis, (hypothalamic-pituitary-thyroid) in the *P. obesus* model. Contrary to the adverse effects observed after T3 administration, 3,5-T2 does not induce any thyrotoxic effect, such as goiter, cardiac, bone or central nervous.

In our study, the detection of TSH in serum could give a more direct indication for feedback effects of 3,5-T_2_. It appears that 3,5-T2 could have a similarly profile and mechanism action that T3 according HPT axis. In addition, this last vision has been revised recently and it seems that 3,5-T2 treatment exerted a negative feedback regulation on the HPT axis, similar to T3, but dose-dependent in 3,5-T2 administration. This is demonstrated by decreased expression of genes responsive to thyroid hormones in pituitary resulting in a suppressed thyroid function with lower T4 and T3 concentrations in serum and liver [42]. Interestingly, in our study, the thyroid-to-body weight ratio increased after 3,5-T2 treatment. Even so, 3,5-T2 treatment down-regulated efficiently T3 levels but maintained in the euthyroid range. Although our data are lacking, the beneficial effects of 3,5-T2 treatment can be explained by the activity of deiodinases [43]. Indeed, 3,5-T2 increases pituitary type 1 iodothyronine deiodinase (D1) activity and transiently decreases type 2 iodothyronine deiodinase (D2) activity [44]. It has been shown that 3,5-T2 binds to activates the human THRβ (Thyroid hormone receptor beta) and exerts genomic effects [45]. Besides, hepatic and kidney deiodinase D1 activities were significantly increased in rats treated with 3,5-T2 despite significantly reduced serum T3 levels, which explains the increase the thyroid-to-body weight ratio in our investigation. As T3 is the main stimulation in liver deiodinase D1 activity through genomic actions and reinforce the suggestion of a T3-like genomic effect of 3,5-T2 [46]. In accordance with this study, others authors have previously showed that 3,5-T2 stimulates D1 activity in rat anterior pituitaries fragments in vivo [47]. Regarding diabetic *P. obesus* (HED-controlled group), it should be noted that T3 plasma levels are increased. Several studies have highlighted the involvement of thyroid hormones in diabetes development [48]. Based on the elevation of T3 levels, it seems that regulation of iodine renal excretion is compromising when the diabetes mellitus is installed [49]. It appears that dysregulation of thyroid function is largely related to the diet. Indeed, previous studies carried on obesity experimental models have shown that a rise in the circulating T3 on the sucrose-rich diet but not with fat-rich diet [50]. Interestingly, in our study, 3,5-T2 treatment decreases significantly T3 plasma levels while T4 and TSH levels remain unchanged. Likewise FT3/FT4 and FT3/3,5-T2-ratio. In our study, the inhibition of T3 secretion by 3,5-T2 treatment appears to be resulted by in rapid elevation circulating plasma levels of 3,5-T2 exogenous (treatment), leading to a feedback effect reducing T3 plasma levels [51]. This explains the decrease in the FT3/FT4, FT3/3,5-T2 and FT4/3,5-T2-ratio.

### 4.2. The Second Point Is Linked to 3,5-T2 Treatment on Body Weight, Calorie Intake, Oxygen Consumption, Respiratory Quotient and Basal Metabolic Rate

According our data, 3,5-T2 treatment allowed weight loss in obese gerbils, concomitantly with stimulation of calorie intake. This result is paradoxical. However, elevated food intake might be a consequence of increase energy expenditure (our study), associated with reduced serum leptin concentrations related to hypothalamic effects after 3,5-T2 treatment [52]; or which might involve altered ghrelin secretion [53]. Indeed, previous studies have shown to depend on hypothalamic activation of the mammalian target of rapamycin pathway (*mTOR* signaling) leading to amplified orexigenic agouti-related peptide, neuropeptide Y, and decreased mRNA levels of anorexigenic proopiomelanocortin in the arcuate nucleus of the hypothalamus [54] representing a strong orexigenic drive. More to this argument, the combination of 3,5-T2 administration and hypercaloric diet (HED) might cooperate at the hypothalamic level via activation of JNK (c-Jun N-terminal kinase) signalling pathway in TRH (thyrotropin-releasing hormone)-secreting neurons, resulting in decreased TRH, TSH, and subsequently thyroid hormone secretion by the thyroid [55]. Regarding oxygen consumption, data from this study showed that during fasting, hepatic oxygen consumption (HOC) is significantly reduced in HED-controlled group compared to the ND-controlled group. Interestingly, 3,5-T2 treatment rapidly stimulated and improved significantly HOC under HED-diet in group III compared to placebo group which increases resting metabolic rate (RMR). Several studies confirm our results. Moreover, this is the first observation that the authors have observed on the effects of 3,5-T2, until then the stimulation of HOC was only attributed to T3 [56]. The HOC enhance can be explained by the hyperactivity of uncoupling proteins linked to thermogenesis regulation by stimulating hepatic fatty acid oxidation inducing a significant increase in mitochondrial respiration (our data on isolated hepatocytes). This leads to a less efficient utilization of lipid substrates, and helps to prevent body-weight gain and hepatic fat accumulation [57]. Relating to respiratory quotient (RQ), in our study we observed that the RQ was between 0.9–1 in the ND and HED-controlled groups which use carbohydrates as energy substrate. However, under 3,5-T2 treatment, the RQ is decreased significantly to 0.7 toward lipids utilization as energy substrate, concomitantly with a significant correlation with the decrease in plasma triglycerides. This indicates the important role of fat metabolism as an energy source in the *P. obesus* model under thyroid hormone action. It is important to note when utilization of carbohydrate is blocked by 3,5-T2 treatment, fat catabolism in the fasted diabetic *P. obesus* (HED-controlled) evolved the same intensity as plasma NEFA (non-esterified fatty acids) and Ketone bodies production (elevated in our study). We observed the same phenomenon with ketogenesis isolated hepatocytes in the presence of octanoate (see below). Concerning basal metabolic rate (BMR) or RMR (resting metabolic rate), our results showed that during fasting, the BMR was significantly lower in ND-group compared to HED-controlled group, indicating greater energy expenditure under HED-diet. However, it is important to emphasize that the BMR of desert rodents is lower than that of non-desert rodents [58]. In our study, the 3,5-T2 treatment increases significantly the BMR associated with fat lipolysis (high NEFA flux). Interestingly, previous study reported an augmented BMR and decreased body weight in obese humans administrated with synthetic analog TRC150094 of 3,5-T2 without sides effects. This action is mediated by T3 action via nuclear TR but with adverse cardiac effects [59]. In this context, recently the mitochondrion is recognized as the major site therapeutic target of 3,5-T2 related to energy expenditure and BMR regulation [60]. Indeed, previous studies were showed a relationship between BMR and stimulation of mitochondrial activity of both cytochrome c-oxidizers and the cytochrome c-reducers components of the respiratory chain, 1 h after the injection of 3,5-T2 [61].

### 4.3. The Third Point Is Linked to 3,5-T2 Treatment on Body Adipose Tissue Distribution

High Energy Diet (HED) in *P. obesus* model (called fat sand rat) has the advantage to mimic most features of human fat overload and overnutrition. HED allows the study of obesity and related adiposity disorders such as abnormalities ectopic fat accumulation on body adipose tissue distribution [62]. Nevertheless, it is important to emphasize that body subcutaneous adipose tissue accretion (but not visceral white depots) protects against systemic glucose intolerance and muscle proteolysis [63], which partly explains the euglycemic state of *P. obesus* in their native desert habitat. According our data, 3,5-T2 treatment is able to prevent and reduce the visceral fat accumulation and to reduced significantly adiposity index. Our results agree with several studies that have tested the beneficial anti-obesity effect of 3,5-T2 treatment. Indeed, Lanni and al. (Goglia laboratory) demonstrated in rats receiving HED + 3,5-T2 (25 µg/100 g body weight) several beneficial effects: body weight loss, stimulation liver fatty acid oxidation, fat mass reduction, disappearance of liver fat (anti-steatotic effect), decreases serum lipids levels (triglycerides, cholesterol) and high adipocyte thermogenesis activity (mitochondrial uncoupling proteins role). The authors involve the stimulation of liver mitochondrial oxidation via the activation of the carnitine palmitoyl-transferase (CPT) system by the AMP-activated protein kinase (AMPK)-acetyl-coenzymeA-carboxylase (ACC)-malonyl CoA signaling pathway [64].

### 4.4. The Fourth Point Is Linked to 3,5-T2 Treatment on Plasma and Liver Metabolic Disorders

In our investigation, we have revealed rise serum 3,5-T2 concentration which can to increase 3,5-T2 in the liver content after 3,5-T2 treatment. This argument indicates a significant hepatic 3,5-T2 uptake by transporter followed by intrahepatic accumulation of 3,5-T2. A similar mechanism could allow the mitochondrial uptake of 3,5-T2 play a role as a reservoir for cytoplasmic 3,5-T2 [65]. The accumulation of 3,5-T2 in the liver could explain in vivo beneficial effects of 3,5-T2, a particular the hypolipidemic, anti-steatotic and anti-hyperglycemic in diabetic *P. obesus*. The accumulation of 3,5-T2 in the liver could be explained by the previous study in rats [66], which reported that 3,5-T2 was more effective in the induction of hepatic expression of the Me1 gene (Malic Enzyme 1) than in circulating TSH suppression. This indicates that tissues and genes were sensitive to the effects of 3,5-T2 are not only linked to the thyroid receptors but also to the cellular availability of the ligands which bind 3,5-T2. Others experimental studies in murine models of obesity have shown that 3,5-T2 treatment is considered as a potential drug for nonalcoholic fatty liver disease (NAFLD) without the detrimental thyromimetic side effects known for classical T3 [67]. The aforementioned beneficial effects of 3,5-T2, the crucial point in this study is linked to attenuation of insulin resistance after 3,5-T2 treatment. This major benefit had favorable repercussions on decline of hyperglycaemia and a hypolipidemic effect. The decrease of hyperinsulinism observed in our study would be explained by an increased insulin catabolism and increase insulin clearance under thyroid hormone action [68]. In agreement with our results, previous studies have shown that 3,5-T2 treatment reduces blood glucose, but independently of insulin sensitization in obese mice. The authors hypothesize this effect through a decrease in hepatic glucose production and changes in energetic substrate utilization [69]. In addition, other authors show that treatment with 3,5-T2 prevents insulin resistance induced by a hypercaloric diet and involve direct rapid effects on mitochondria in target tissues, enhanced AMPK activity and activation of hepatic nuclear sirtuin 1 [70]. Besides, our results can be argued by the inhibition of isolated hepatocytes gluconeogenesis as lactate substrate (see below), which explains the attenuation of fasting hyperglycemia. In addition, to explain the attenuation of hyperglycemia, some studies conducted in obesity model reveal a stimulation of the oxidation of glucose via the activation hepatic mitochondrial L-glycerol 3 phosphate dehydrogenase [71], concomitantly with increased the activity of hepatic mitochondrial FAD-glycerophosphate oxidase [72]. On the other hand, we showed that 3,5-T2 treatment allowed to decrease the hepatic glycogen content. This result can be explained by two possibilities, either by an inhibiting effect on glycogen synthesis or by stimulation of glycogenolysis to supply glucose tissues for energy expenditure. The literature describes that these two metabolic pathways seem to be controlled by two signaling pathways: AKT (protein B kinase) or GSK-3 (glycogen synthase kinase-3) phosphorylation which is incriminated by 3,5-T2 inhibition action [73]. Interestingly, 3,5-T2 strongly reduced hepatic expression of GLUT2, which mediates bidirectional transport of glucose across the hepatocyte plasma membrane, and decreased hepatic glucose output. Hepatic GLUT2 expression is commonly up-regulated in obese and diabetic animals and high hepatic GLUT2 expression may contribute to hyperglycemia [74].

### 4.5. The Fifth Point Is Linked to 3,5-T2 Treatment on Hepatic Gluconeogenesis, Ketogenesis, Intracellular Intermediary Metabolites, and Cellular Redox-Phosphate Potential

Our data strongly implicate the liver as a major of this finding related to 3,5-T2 effects. We have investigated if 3,5-T2 can regulate metabolic homeostasis in hepatocytes focusing on the metabolic mechanisms involved. In our study, we have demonstrated that 3,5-T2 treatment at 10^−6^ M decreased significantly hepatic glucose output from both gluconeogenic substrates conditions (alanine + octanoate or lactate + octanoate). We deduce, an action anti-gluconeogenesis and anti-ketogenesis of 3,5-T2. Several in vitro studies have investigated whether 3,5-T2 directly acts on the liver apart from metabolic pathway involvement by exposing rat hepatocytes to a mixture of substrates and treating them with various doses of 3,5-T2 [75,76]. Recent studies have shown that 3,5-T2 stimulate the mTORC2/Akt complex. This pathway controls the insulin-induced inhibition of gluconeogenesis via an AMPK-mediated mechanism, leading to the downregulation of the key expression gluconeogenic genes in hepatocytes [77]. In the present study, 3,5-T2 at a concentration of at 10^−6^ M stimulated ATP-consuming processes gluconeogenesis as well as adenine nucleotide (NAD) translocation from mitochondria to cytosol. Concomitantly, the effects of 3,5-T2 on cytosolic and mitochondrial redox potentials have an impact on cytosolic and mitochondrial ATP production and phosphate potential (ATP/ADP). Under 3,5-T2 treatment, the ATP regeneration much more from lactate than from alanine as an energetic substrate, which is associated with an acceleration of mitochondrial adenine nucleotide transport. The treatment by 3,5-T2 allows to increase significantly the cytosolic ATP/ADP ratio, but decrease the mitochondrial ATP/ADP ratio. Our results agree with previous studies. Authors have demonstrated that 3,5-T2 acts rapidly and directly via a mitochondrial pathway, whereas T3 exerts its long-term action indirectly by induction of specific enzymes [78]. These data have recently been confirmed by other studies [79]. Regarding redox potential data, we showed that 3,5-T2 treatment decreases cytosolic redox potential (L/P ratio) and increases mitochondrial (3-βOHB/AcAc ratio). The 3-βOHB/AcAc ratio is used as a surrogate for mitochondrial redox potential of ketone bodies, was consistent with the increases in oxygen consumption observed in the in vitro study. In addition, 3,5-T2 treatment increased hepatic mitochondrial fatty acid oxidation and CPT activity [80]. However, in our study, it seems that the activation of substrate oxidation (alanine, glucose, pyruvate, lactate, octanoate) would not be coupled to mitochondrial phosphorylation, because the mitochondrial ATP/ADP ratio is decreased. The most probable explanation would be related to the process of the effect of 3,5-T2 on uncoupled respiration involve the mechanisms proton leak and slipping are more important under the 3,5-T2 treatment [81]. Indeed, it has long been known that ATP synthesis is not perfectly coupled to mitochondrial respiration. Since protons can leak across the inner membrane without contributing to the ATP synthesis, the energy of the proton electrochemical gradient is released as heat. Mitochondrial proton leak (or uncoupling of mitochondria), which in isolated hepatocytes accounts for 20–30% of oxygen consumption, is known to be affected by thyroid hormones [82].

## 5. Conclusions

Based our experimental findings, the therapeutic strategy via the 3,5-T2 treatment without causing hyperthyroidism and side adverse seems showed a promising future for the obesity and the prevention of its metabolic complications, particularly in type 2 diabetes. Our in vivo and in vitro data confirmed its effectiveness to ameliorate ketoacidosis and lipid metabolism in diabetic *P. obesus*; these events likely relying on hepatocellular redox status and mitochondrial activity. This study likewise agreed with the successful use of 3,5-T2 for patients obese, intolerant to glucose or with moderate hyperglycemia. Such data are of utmost importance for innovative therapeutic strategies, and any comparison between the natural history of diabetes in *P. obesus* and human pathology should help explain, at least partially, the higher risk of developing morbid obesity for many people with unhealthy dietary habits.

## Figures and Tables

**Figure 2 nutrients-14-03044-f002:**
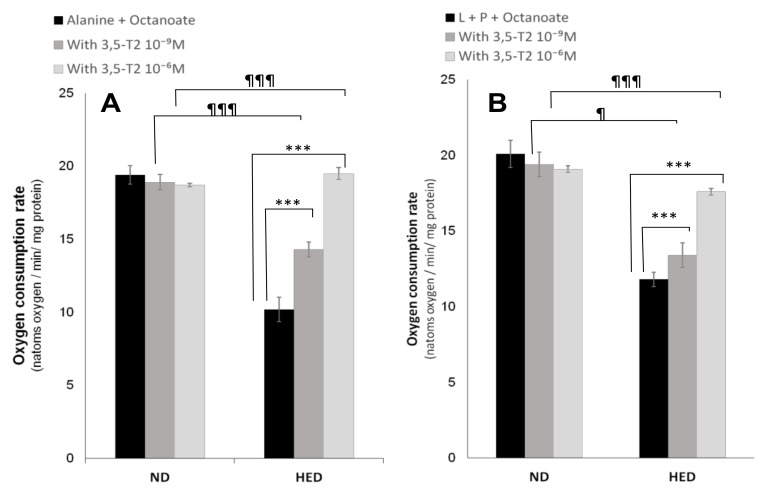
Effects of 3,5-T2 on oxygen consumption in isolated hepatocytes from experimental *P. obesus* fed ND or HED. ND: Natural Diet, HED: High Energy Diet. Freshly isolated hepatocytes from 16 h-fasted *P. obesus* were incubated in Krebs/bicarbonate, buffer containing alanine or lactate (L) + pyruvate (P) together with octanoate (**A**,**B**) in the absence or presence of 3,5-T2 (10^−9^ or 10^−6^ M). The oxygen consumption rate was determined polarographically at 37 °C with a Clark electrode and expressed as natoms O_2_/min/mg protein. Data are expressed as mean ± SEM. *** *p* < 0.001 compared 3,5-T2-treated group versus HED placebo group. ^¶^ p < 0.05; ^¶¶¶^ p < 0.001 compared 3,5-T2-treated group versus ND placebo group. ns: no significant.

**Figure 3 nutrients-14-03044-f003:**
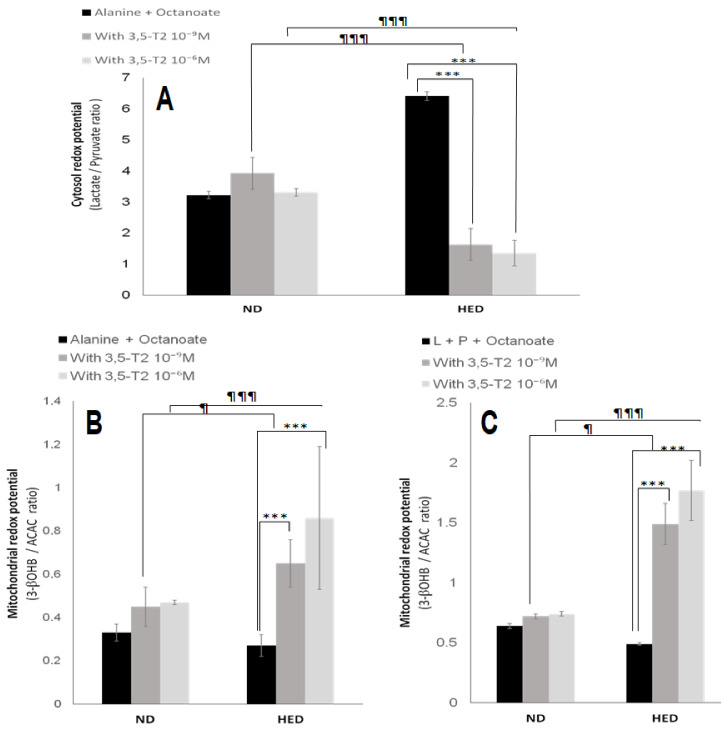
Effects of 3,5-T2 on both cytosolic and mitochondrial redox potential (NADH/NAD^+^) in isolated hepatocytes from experimental *P. obesus* fed ND or HED. ND: Natural Diet, HED: High Energy Diet. Freshly isolated hepatocytes from 16 h-fasted *P. obesus* were incubated in Krebs/bicarbonate, buffer containing alanine or lactate (L) + pyruvate (P) together with octanoate in the absence or presence of 3,5-T2 (10^−9^ or 10^−6^ M). Cytosol redox potential is represented by the L/P ratio (**A**). Mitochondrial redox potential is represented by the β- hydroxybutyrate/acetoacetate ratio (β-HB/AcAc) (**B**,**C**). Data are expressed as mean ± SEM. *** *p* < 0.001 compared 3,5-T2-treated group versus HED placebo group. ^¶^ *p* < 0.05; ^¶¶¶^ *p* < 0.001 compared 3,5-T2-treated group versus ND placebo group. ns: no significant.

**Figure 4 nutrients-14-03044-f004:**
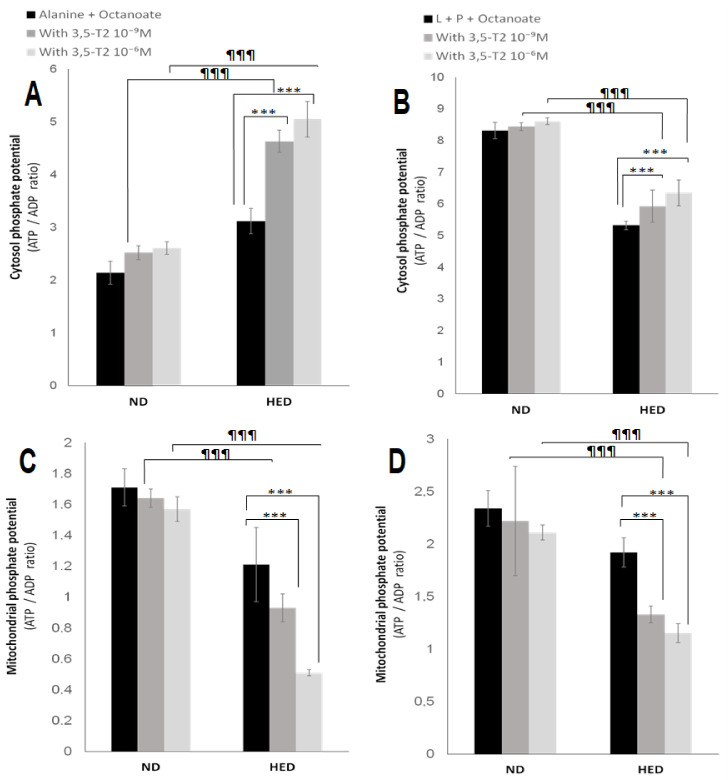
Effects of 3,5-T2 on both cytosolic and mitochondrial phosphate potential (ATP/ADP) in isolated hepatocytes from experimental *P. obesus* fed ND or HED. ND: Natural Diet, HED: High Energy Diet. Freshly isolated hepatocytes from 16 h-fasted *P. obesus* were incubated in Krebs/bicarbonate, buffer containing alanine or lactate (L) + pyruvate (P) together with octanoate in the absence or presence of 3,5-T2 (10^−9^ or 10^−6^ M). Cytosol (**A**,**B**) and mitochondrial (**C**,**D**) phosphate potential is represented by the ATP/ADP ratio. Data are expressed as mean ± SEM. *** *p* < 0.001 compared 3,5-T2-treated group versus HED placebo group. ^¶¶¶^ *p* < 0.001 compared 3,5-T2-treated group versus ND placebo group. ns: no significant.

**Table 1 nutrients-14-03044-t001:** In vivo effects of 3,5-T2 on thyroid hormones plasma levels from experimental *P. obesus*.

Parameters/Groups	Group I	Group II	Group III	Group IV
	ND-controlled	HED-controlled	HED-3,5-T2-treated	HED-Placebo
Body weight (g)	81 ± 3	144 ± 10 ***	108 ± 3 ***^/¶¶¶^	141 ± 7 ***
Thyroid weight (mg)	11.5 ± 1.32	10.7 ± 2.69 ^ns^	10.3 ± 1.18 ^ns/ns^	10.1 ± 1.47 ^ns^
Thyroid mass (% body wt)	14.1 ± 0.44	7.43 ± 0.26 ***	9.53 ± 0.39 ***^/¶¶¶^	7.16 ± 0.21 ***
TSH (µIU/mL)	1.63 ± 0.52	1.37 ± 0.17 ^ns^	1.19 ± 0.49 ^ns/ns^	1.46 ± 0.81 ^ns^
TT4 (ng/mL)	29.4 ± 2.23	24.3 ± 4.11 ^ns^	23.9 ± 3.22 ^ns/ns^	23.7 ± 1.34 ^ns^
FT4 (pg/mL)	3.88 ± 0.11	3.75 ± 0.57 ^ns^	3.71 ± 0.66 ^ns/ns^	3.79 ± 0.45 ^ns^
TT3 (ng/mL)	0.661 ± 0.08	0.344 ± 0.07 ***	0.362 ± 0.03 ***^/ns^	0.377 ± 0.09 ***
FT3 (pg/mL)	1.92 ± 0.08	1.77 ± 0.07 **	1.10 ± 0.09 ***^/¶¶¶^	1.83 ± 0.02 **
3,5-T2 (pg/mL)	0.30 ± 0.09	0.24 ± 0.07 ^ns^	0.51 ± 0.05 ***^/¶¶¶^	0.27 ± 0.08 ^ns^
FT3/FT4-ratio	0.49 ± 0.07	0.47 ± 0.01 ^ns^	0.29 ± 0.03 ***^/¶¶¶^	0.48 ± 0.04 ^ns^
FT3/3,5-T2-ratio	6.41 ± 0.88	7.37 ± 0.11 ^ns^	2.15 ± 0.18 ***^/¶¶¶^	6.77 ± 0.25 ^ns^
FT4/3,5-T2-ratio	12.9 ± 1.22	15.6 ± 1.42 **	7.27 ± 1.32 ***^/¶¶¶^	14.1 ± 0.11 **

Groups: I (n = 32); II (n = 30); III (n = 65) and IV (n = 30). ND: Natural Diet; HED: High Energy Diet; TSH: Thyreostimulating hormone; TT4: total Thyroxine; TT3: total 3,5,3′-triiodothyronine; FT4: Free Thyroxine or 3,5,3′,5′-tetraiodothyronine; FT3: Free 3,5,3′-triiodothyronine; 3,5-T2: 3,5-diiodothyronine. Normal range are as follows: TSH = 0.25–4 µIU/mL; TT4 = 45–126 ng/mL; FT4 = 7–18 pg/mL; TT3 = 0.80–2.20 ng/mL; FT3 = 2.2–5.3 pg/mL; 3,5-T2: 1.5 ± 4.6 pg/mL; FT3/FT4-ratio = 1.42–3.05. Normal range of FT3/3,5-T2-ratio and FT4/3,5-T2-ratio were not defined. Plasma thyroid hormones were assayed in 16 h-fasted *P. obesus*. Data are expressed as mean ± SEM. ** *p* < 0.01; *** *p* < 0.001 compared 3,5-T2-treated group to HED placebo group. ^¶¶¶^ *p* < 0.001 compared 3,5-T2-treated group to ND placebo group. ns: no significant.

**Table 2 nutrients-14-03044-t002:** In vivo effects of 3,5-T2 on resting energy expenditure and plasma metabolic biomarkers from experimental *P. obesus*.

Parameters/Groups	Group I	Group II	Group III	Group IV
	ND-controlled	HED-controlled	HED-3,5-T2-treated	HED-Placebo
Body weight (g)	81 ± 3	144 ± 10 ***	108 ± 3 ***^/¶¶¶^	141 ± 7 ***
BMI (g/cm^2^)	0.38 ± 0.01	0.49 ± 0.01 ***	0.32 ± 0.03 ^ns/¶¶¶^	0.47 ± 0.02 ***
Caloric intake (Kcal/100g BW)	43.5 ± 1.9	316 ± 11 ***	534 ± 10 ***^/¶¶¶^	322 ± 17 ***
Respiratory quotient	0.913 ± 0.09	0.975 ± 0.03 ^ns^	0.753 ± 0.05 ***^/¶¶¶^	0.944 ± 0.07 ^ns^
BMR (mLO_2_/h/g BW)	0.521 ± 0.02	0.643 ± 0.09 ***	0.891 ± 0.05 ***^/¶¶¶^	0.675 ± 0.03 ***
Glucose (mmol/L)	3.21 ± 0.51	15.2 ± 1.04 ***	3.88 ± 0.17 ^ns/¶¶¶^	14.9 ± 1.82 ***
HbA1c (mmol/mol)	17.9 ± 0.65	68.2 ± 9.44 ***	20.5 ± 3.85 ^ns/¶¶¶^	65.3 ± 7.11 ***
Insulin (pmol/L)	130 ± 21	580 ± 47 ***	110 ± 14 ***^/¶¶¶^	608 ± 85 ***
HOMA-IR	2.57 ± 0.14	5.33 ± 0.23 ***	2.68 ± 0.11 ^ns/¶¶¶^	5.58 ± 0.44 ***
ALT (IU/L)	26.3 ± 1.58	74.1 ± 2.35 ***	23.9 ± 1.47 ^ns/¶¶¶^	81.3 ± 1.71 ***
Triglycerides (mmol/L)	0.81 ± 0.07	4.09 ± 0.61 ***	1.58 ± 0.01 ***^/¶^	3.07 ± 0.06 ***
Total cholesterol (mmol/L)	1.48 ± 0.70	8.06 ± 1.22 ***	2.81 ± 0.64 ***^/¶¶¶^	7.98 ± 1.15 ***
NEFA (μmol/L)	291 ± 33	578 ± 41 ***	898 ± 66 ***^/¶¶¶^	609 ± 23 ***
Ketone bodies (µmol/L)	250 ± 16	303 ± 30 ***	588 ± 57 ***^/¶¶¶^	278 ± 21 ***
Lactate (mM)	0.65 ± 0.09	1.13 ± 0.25 ***	0.73 ± 0.14 ***^/¶¶¶^	1.05 ± 0.03 ***

Groups: I (n = 32); II (n = 30); III (n = 65) and IV (n = 30). ND: Natural Diet; HED: High Energy Diet; BW: body weight; BMR: basal metabolic rate. BMI: body mass index; HbA1c: Glycated hemoglobin; HOMA-IR: Homeostatic Model Assessment-Insulin Resistance; ALT: alanine aminotransferase; NEFA: non-esterified fatty acids. Data are expressed as mean ± SEM. *** *p* < 0.001 compared 3,5-T2-treated group versus HED placebo group. ^¶^ *p* < 0.05; ^¶¶¶^ *p* < 0.001 compared 3,5-T2-treated group versus ND placebo group. ns: no significant. Ketone bodies are représented by the somme of β- hydroxybutyrate and acetoacetate.

**Table 3 nutrients-14-03044-t003:** In vivo effects of 3,5-T2 on body adipose tissue repartition and hepatic metabolic biomarkers from experimental *P. obesus*.

Parameters/Groups	Group I	Group II	Group III	Group IV
	ND-controlled	HED-controlled	HED-3,5-T2-treated	HED-Placebo
Total visceral fat (mg/g BW)	10.3 ± 3.38	25.8 ± 4.32 ***	10.1 ± 1.85 ***^/¶¶¶^	26.3 ± 4.38 ***
Retroperitoneal fat (mg/g BW)	2.75 ± 0.66	3.02 ± 0.57 ^ns^	2.11 ± 0.44 ^ns/ns^	2.98 ± 0.75 ^ns^
Omental fat (mg/g BW)	1.27 ± 0.91	9.95 ± 1.33 ***	2.79 ± 0.68***^/¶¶¶^	10.1 ± 2.22 ***
Mesenteric fat (mg/g BW)	6.33 ± 1.81	12.9 ± 2.42 ***	5.18 ± 0.73 ***^/¶¶¶^	13.2 ± 1.41 ***
Epididymal fat (mg/g BW)	3.14 ± 0.72	8.69 ± 1.47 ***	2.25 ± 0.65 ***	9.02 ± 0.73 ***
Subcutaneous fat (mg/g BW)	1.93 ± 0.17	8.33 ± 1.18 ***	2.09 ± 0.15 ***^/¶¶¶^	7.92 ± 0.22 ***
Perirenal fat (mg/g BW)	2.13 ± 0.51	2.82 ± 1.61 ^ns^	2.03 ± 0.34 ^ns/ns^	2.57 ± 1.91 ^ns^
Suprascapular fat (mg/g BW)	5.91 ± 2.14	12.5 ± 1.08 **	4.11 ± 1.33 ***^/¶¶¶^	11.9 ± 2.01 **
Gonadal fat (mg/g BW)	1.78 ± 0.61	1.93 ± 0.22 ^ns^	1.66 ± 0.55 ^ns/ns^	1.86 ± 1.24 ^ns^
Intramuscular fat (mg/g BW)	2.51 ± 0.33	3.74 ± 0.27 **	1.81 ± 0.24 ***^/¶¶¶^	3.67 ± 0.45 **
Adiposity index (% BW)	2.61 ± 0.35	3.87 ± 0.95 ***	1.07 ± 0.13 **^/¶¶¶^	3.91 ± 0.47 ***
Brown adipose tissue (mg/g BW)	0.46 ± 0.03	0.33 ± 0.07 *	0.55 ± 0.05 *^/¶^	0.31 ± 0.02 *
Total hepatic lipids (g 100 g wet/wt)	2.97 ± 0.31	6.17 ± 0.94 ***	3.19 ± 0.11 ***^/¶¶¶^	5.83 ± 0.76 ***
Hepatic glycogen (g/100 g wet/wt)	2.77 ± 0.84	2.44 ± 0.93 *	1.18 ± 0.02 ***^/¶¶¶^	2.65 ± 0.79
Liver mass (% body BW)	2.51 ± 0.45	4.05 ± 0.78 ***	2.89 ± 0.33 ***^/¶¶¶^	3.64 ± 1.49 ***

Groups: I (n = 32); II (n = 30); III (n = 65) and IV (n = 30). ND: Natural Diet; HED: High Energy Diet; BW: body weight. wt: weight. Data are expressed as mean ± SEM. * *p* < 0.05; ** *p* < 0.01; *** *p* < 0.001 compared 3,5-T2-treated group versus HED placebo group. ^¶^
*p* < 0.05; ^¶¶¶^
*p* < 0.001 compared 3,5-T2-treated group versus ND placebo group. ns: no significant.

**Table 4 nutrients-14-03044-t004:** In vitro effects of 3,5-T2 on hepatic gluconeogenesis, glycolysis, ketogenesis and intracellular intermediary metabolites fluxes from experimental *P. obesus*.

Metabolic Fluxes (µmol/min/g Dry Cells)	Energy Substrates (mM)	Group I ND-Controlled	Group II HED-Controlled	Group III HED-3,5-T2-Treated		Group IV HED-Placebo	
				**10^−9^ M** 3,5-T2	**10^−6^ M** 3,5-T2	**10^−9^ M** 3,5-T2	**10^−6^ M** 3,5-T2
Glucose synthesis	Ala+Octa	4.33 ± 0.22	8.38 ± 0.54	2.55 ± 0.31 ***^/¶¶¶^	2.19 ± 0.11 ***^/¶¶¶^	4.22 ± 0.62	3.87 ± 0.22
	(L+P)+Octa	5.66 ± 0.31	12.1 ± 0.27	4.87 ± 0.17 ***^/¶^	2.66 ± 0.33 ***^/¶¶¶^	5.33 ± 0.41	5.02 ± 0.77
Glycolysis	Ala+Octa	1.83 ± 0.12	3.27 ± 0.14	1.65 ± 0.41 ***^/¶¶¶^	1.09 ± 0.53 ***^/¶¶¶^	2.02 ± 0.33	2.11 ± 0.12
Ketogenesis	Ala+Octa	4.77 ± 0.33	5.65 ± 0.61	5.87 ± 0.55 ***^/¶¶^	6.29 ± 0.11 ***^/¶¶¶^	4.08 ± 0.34	3.95 ± 0.61
	(L+P)+Octa	3.76 ± 0.51	4.61 ± 0.37	4.97 ± 0.11 ***^/¶^	5.44 ± 0.13 ***^/¶¶¶^	3.05 ± 0.21	3.02 ± 0.17
G6P (nmol/g dry cells)		184 ± 62	227 ± 51	108 ± 31 ***^/¶¶¶^	97 ± 14 ***^/¶¶¶^	172 ± 23	181 ± 55
F6P (nmol/g dry cells)		299 ± 24	338 ± 55	317 ± 11 ***^/¶¶¶^	379 ± 19 ***^/¶¶¶^	304 ± 31	310 ± 22
3PG (nmol/g dry cells)		951 ± 16	1331 ± 77	1016 ± 33 ***^/¶¶¶^	1287 ± 46 ***^/¶¶¶^	997 ± 32	1008 ± 57
PEP (μmol/g dry cells)		575 ± 30	805 ± 44	895 ± 41 ***^/¶¶¶^	1034 ± 27 ***^/¶¶¶^	609 ± 22	645 ± 18

Groups: I (n = 32); II (n = 30); III (n = 65) and IV (n = 30). ND: Natural Diet; HED: High Energy Diet; Freshly isolated hepatocytes from 16 h-fasted *P. obesus* were incubated in Krebs/bicarbonate, buffer containing Alanine (Ala) or Lactate (L) + Pyruvate (P) together with Octanoate (Octa), in the absence or presence of 3,5-T2 (10^−9^ or 10^−6^ M). G6P: Glucose 6-phosphate, F6P: Fructose 6-phosphate, PEP: phosphoenolpyruvate, 3PG: 3-phosphoglycerate are procuded from L + P + Octa. Data are expressed as mean ± SEM. *** *p* < 0.001 compared 3,5-T2-treated group versus HED placebo group. ^¶^ *p* < 0.05; ^¶¶^ *p* < 0.01; ^¶¶¶^ *p* < 0.001 compared 3,5-T2-treated group versus ND placebo group. ns: no significant.

## Data Availability

The data presented in this study are available on request from the corresponding author.

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
