# Peer review of "Potential Applications of Thyroid Hormone Derivatives in Obesity and Type 2 Diabetes: Focus on 3,5-Diiodothyronine (3,5-T2) in Psammomys obesus (Fat Sand Rat) Model"

_nutrients, 2022, doi:10.3390/nu14153044_

Round 1
Reviewer 1 Report
This study examined 3,5-T2 in the prevention of energy metabolism disorders in diabetes-prone P. obesus gerbils. 3,5-T2 was tested at a dose of 50 μg and was administered subcutaneously. Treatment with 3,5-T2 reduces visceral adipose tissue, prevents insulin resistance, attenuated hyperglycemia, dyslipidemia and hepatic steatosis. 3,5-T2 reduced both gluconeogenesis and ketogenesis.
The subject is very complex but the experiment was conducted with precision and the results, even if taken for granted, are very good.
The phrase that states: "the use of 3,5-T2 as a natural therapeutic means to regulate cellular energy metabolism may be recommended." as the use on humans is still far away.
Reviewer 2 Report
This is a randomized prospective study that looks at the potential use of 3,5 T2 as a therapeutic means for metabolic disturbances associated with type 2 diabetes and obesity. The therapeutic strategy is based on the non-genomic actions of thyroid hormone derivates (THD) that are mediated by their mitochondrial or cell membrane binding without causing hyperthyroidism.
The manuscript is clear, comprehensive, relevant to the field, and presented in a well-structured manner. It is sound scientifically, and the experimental design is appropriate.
The introduction is short, informative, sufficient, and gives an overview of the mechanisms of THD actions.
The methods are written and explained in great detail, and the results could be reproducible based on them.
In the study, the gerbil Psammomys obesus was used as a spontaneous relevant polygenic human model for nutritionally induced obesity, non-alcoholic fatty liver disease, type 2 diabetes mellitus, and atherosclerosis disease. They were first randomized into two diet-experimental groups (natural diet-ND and High-Energy-Diet-HED), and then the HED-fed group was randomized into three subgroups (HED control, HED 3, 5-T2-treated group, and HED placebo-controlled group) that are clearly described in Figure 1.
All the experimental procedures were authorized by the relevant committee, and the permits and ethical rules have been obtained, which are clearly stated in the manuscript.
The results are clearly presented in five subsections according to the in-vivo and in-vitro effects of 3,5-T2, and are shown in four tables and three figures. The tables and figures show the data properly and are easy to interpret and understand. The results are interpreted appropriately, consistently, and clearly throughout the manuscript. The statistical analysis is clear.
The results showed a significant beneficial effect of 3,5-T2 treatment on body weight (reduced), respiratory quotient (decreased) and basal metabolic rate (increased), while the calorie intake was increased. The treatment with 3,5 T2 also showed a significant decrease in metabolic parameters such as glucose, HbA1c, insulin, HOMA IR, trigliceryde, total cholesterol, lactate, and ALT as a marker of hepatocyte damage, while increasing NFA and ketone bodies. Adiposity index was reduced significantly and there was a positive impact on visceral and subscapular adipose tissue mass reduction, while brown adipose tissue was increased under 3,5-T2 treatment. There was also a reduction in total hepatic lipids, hepatic glycogen, and liver mass. In vitro studies on freshly isolated hepatocytes showed that 3,5-T2 treatment increased oxygen consumption, decreased hepatic glucose output, glucose synthesis, and glycolysis in a dose-dependent manner, but only in HED-fed gerbils. The treatment by 3,5-T2 also increased significantly the cytosolic ATP/ADP ratio while decreasing significantly the mitochondrial phosphate potential, i.e., the mitochondrial ATP/ADP ratio and decreased cytosolic while increasing mitochondrial redox potential. It was also shown that 3,5-T2 treatment significantly increased phosphoenolpyruvate, 3-phosphoglycerate, and fructose 6-phosphate levels as gluconeogenic substrates, while greatly decreasing glucose 6-phosphate levels.
The results are consistent with the results from previous similar studies.
The discussion is of an informative nature; it discusses the research results in the light of previous research in the same area with a critical review of the possible systemic effects of THD, i.e., the impact on the HPT axis or genomic action of THD via nuclear thyroid hormone receptors. This study shows that the HPT axis was not disturbed in these gerbils under 3,5-T2 treatment, and that a thyrotoxic state was not induced.
The conclusions are consistent with the evidence and arguments presented. The authors concluded that the therapeutic strategy via the 3,5-T2 treatment without causing hyperthyroidism and side effects showed a promising future for obesity and the prevention of its metabolic complications, particularly in type 2 diabetes.
The cited references are mostly recent publications and relevant to the topic.
Minor changes in English grammar and style are needed. For example,
548-554-rephrase the sentences. It is not written comprehensibly.
688- 689 This observation possibly reveals that 3,5-T2 treatment would be able to inhibit (s) glycogen synthesis or to activate glycogenolysis.
690-691 The sentence is not understandable.
Reviewer 3 Report
Dear Authors,
The manuscript entitled “Potential applications of Thyroid hormone derivates in obesity and type 2 diabetes: Focus on 3,5-diiodothyronine (3,5-T2) in Psammomys obesus (fat sand rat) model” offers a very exhaustive experimental study.
The aim of this work was to analyze the potential therapeutic effect of thyroid hormones derivatives (THD) through non-nuclear mechanisms, which can be effective in cases of obesity and/or type 2 diabetes, avoiding the appearance of hyperthyroidism. The pleiotropic effects of 3,5-T2 on thyroid function as well as glucose metabolism, adipose tissue regulation (body weight, calorie intake, basal metabolic rate), oxygen consumption and respiratory quotient has been measured thoroughly.
The manuscript is brilliantly written, the methods have been described in detail and the results can be easily understood thank to tables and figures. The discussion is certainly valuable and contains plausible explanations for the findings.
I have just one tiny remark:
Lines 749-753: Since the microbiota of the gastrointestinal tract has not studied, it does not seem correct to include these sentences as conclusion.
Author Response
Point-by-point rebuttals / answers to the reviewer’s comments
Manuscript ID: Nutrients -1830573 - Minor Revisions
Date of submission: 07/07/2022 - Date of the reply to the submission: 17/07/2022 by the journal
Revised file within 05 days (23/07/2022)
Reviewer # 3
____________________________________
Title: Potential applications of Thyroid hormone derivatives in obesity and type 2 diabetes: Focus on 3, 5-diiodothyronine (3, 5 -T2) in Psammomys obesus (fat sand rat) model
Asma Bouazza1, Roland Favier2, Eric Fontaine2, Xavier Leverve2 and Elhadj-Ahmed Koceir1,*
1 Nutrition and Dietetics in Human Pathologies Post Graduate School, Bioenergetics and Intermediary Metabolism team, Biology and Organisms Physiology laboratory, USTHB, El Alia, Bab Ezzouar, 16123, Algiers, Algeria; bouazza.asma@gmail.com
2 Laboratory of Fundamental and Applied Bioenergetics (LBFA), INSERM, U1055, Grenoble, France; roland.favier@univ-grenoble-alpes.fr ; eric.fontaine@univ-grenoble-alpes.fr; xavier.leverve@univ-grenoble-alpes.fr
* Correspondence: e.koceir@gmail.com; Tel (personal): 213. (0)6. 66.74.27.70; Fax/Tel (USTHB): 213. (0)21.24.72.17
General Comments
. I would like to sign my review report
. English language and style: I don't feel qualified to judge about the English language and style
- Does the introduction provide sufficient background and include all relevant references? Yes
- Are all the cited references relevant to the research? Yes
- Is the research design appropriate? Yes
- Are the methods adequately described? Yes
- Are the results clearly presented? Yes
- Are the conclusions supported by the results? Yes
The manuscript entitled “Potential applications of Thyroid hormone derivates in obesity and type 2 diabetes: Focus on 3,5-diiodothyronine (3,5-T2) in Psammomys obesus (fat sand rat) model” offers a very exhaustive experimental study.
The aim of this work was to analyze the potential therapeutic effect of thyroid hormones derivatives (THD) through non-nuclear mechanisms, which can be effective in cases of obesity and/or type 2 diabetes, avoiding the appearance of hyperthyroidism. The pleiotropic effects of 3,5-T2 on thyroid function as well as glucose metabolism, adipose tissue regulation (body weight, calorie intake, basal metabolic rate), oxygen consumption and respiratory quotient has been measured thoroughly. The manuscript is brilliantly written, the methods have been described in detail and the results can be easily understood thank to tables and figures. The discussion is certainly valuable and contains plausible explanations for the findings.
Thank you very much for having evaluated our study favorably with a very relevant scientific analysis, underlining your notoriety in this thyroid hormones research field.
Comments and Suggestions for Authors
I have just one tiny remark:
Lines 749-753: Since the microbiota of the gastrointestinal tract has not studied, it does not seem correct to include these sentences as conclusion.
We appreciate and fully agree with this comment. As suggested, we corrected the conclusion. We have deleted the paragraph: “On the over hand, part of 3,5-T2 might directly interact with the microbiota of gastrointestinal tract before entering the circulation and peripheral tissues to protect host cells against severe damages. This open question linked to the dependency and interaction of 3,5-T2 effects with microbiome in metabolic diseases needs to be further elucidated” Please see the Track changes Manuscript (MS).